# PERMUTATION EQUIVARIANT MODELS FOR COMPOSITIONAL GENERALIZATION IN LANGUAGE

**Jonathan Gordon***
University of Cambridge
jg801@cam.ac.uk

**David Lopez-Paz, Marco Baroni, Diane Bouchacourt**
Facebook AI Research
{dlp, mbaroni, dianeb}@fb.com

## ABSTRACT

Humans understand novel sentences by composing meanings and roles of core language components. In contrast, neural network models for natural language modeling fail when such compositional generalization is required. The main contribution of this paper is to hypothesize that language compositionality is a form of group-equivariance. Based on this hypothesis, we propose a set of tools for constructing equivariant sequence-to-sequence models. Throughout a variety of experiments on the SCAN tasks, we analyze the behavior of existing models under the lens of equivariance, and demonstrate that our equivariant architecture is able to achieve the type compositional generalization required in human language understanding.

## 1 INTRODUCTION

When using language, humans recombine known concepts to understand novel sentences. For instance, if one understands the meaning of "run", "jump", and "jump twice", then one understands the meaning of "run twice", even if such sentence was never heard before. This relies on the notion of *language compositionality*, which states that the meaning of a sentence ("jump twice") is to be obtained by the meaning of its constituents (e.g. the verb "jump" and the quantifying adverb "twice") and the use of algebraic computation (a verb combined with a quantifying adverb $m$ results in doing that verb $m$ times) (Kratzer & Heim, 1998).

In the realm of machines, deep learning has achieved unprecedented results in language modeling tasks (Bahdanau et al., 2015; Vaswani et al., 2017). However, these models are sample inefficient, and do not generalize to examples that require the use of language compositionality (Lake & Baroni, 2018; Loula et al., 2018; Dessì & Baroni, 2019). This result suggests that deep language models fail to leverage compositionality; a failure remaining to this day a roadblock towards true natural language understanding.

Focusing on this issue, Lake & Baroni (2018) proposed the Simplified version of the CommAI Navigation (SCAN), a dataset to benchmark the compositional generalization capabilities of state-of-the-art sequence-to-sequence (seq2seq) translation models (Sutskever et al., 2014; Bahdanau et al., 2015). In a nutshell, the SCAN dataset contains compositional navigation commands such as JUMP TWICE AFTER RUN LEFT, to be translated into the sequence of actions LTURN RUN JUMP JUMP.

Using SCAN, Lake & Baroni (2018) demonstrated that seq2seq models fail spectacularly at tasks requiring the use of language compositionality. Following our introductory example, models trained on the three commands JUMP, RUN and JUMP TWICE fail to generalize to RUN TWICE. Most recently, Dessì & Baroni (2019) showed that architectures based on temporal convolutions meet the same fate.

SCAN did not only reveal the lack of compositionality in language models, but it also became the blueprint to build novel language models able to handle language compositionality. On the one hand, Russin et al. (2019) proposed a seq2seq model where semantic and syntactic information are represented separately, in a hope that such disentanglement would elicit compositional rules. However, their model was not able to solve all of the compositional tasks comprising SCAN. On the other hand, Lake (2019) introduced a meta-learning approach with excellent performance in multiple

---

*Work conducted during an internship at Facebook AI Research

SCAN tasks. However, their method requires substantial amounts of additional supervision, and a complex meta-learning procedure hand-engineered for each task.

In this paper, we take a holistic look at the problem and connect language compositionality in SCAN to the disparate literature in models *equivariant* to certain *group symmetries* (Kondor, 2008; Cohen & Welling, 2016; Ravanbakhsh et al., 2017; Kondor & Trivedi, 2018). Interesting links have recently been proposed between group symmetries and the areas of causality (Arjovsky et al., 2019) and disentangled representation learning (Higgins et al., 2018), and this work proceeds in a similar fashion. In particular, the **main contribution** of this work is not to chase performance numbers, but to put forward the novel hypothesis that *language compositionality can be understood as a form of group-equivariance* (Section 3). To sustain our hypothesis, we provide tools to construct seq2seq models equivariant when the group symmetries are known (Section 4), and demonstrate that these models solve all SCAN tasks, except length generalization (Section 6).[1]

## 2 THE SCAN COMPOSITIONAL TASKS

The purpose of the Simplified version of the CommAI Navigation (SCAN) tasks (Lake & Baroni, 2018) is to benchmark the abilities of machine translation models for compositional generalization. Following prior literature (Lake & Baroni, 2018; Baroni, 2019; Russin et al., 2019; Andreas, 2019), compositional generalization is understood as the ability to translate novel families of sentences, when this requires leveraging the compositional structure in language.

The SCAN dataset contains compositional navigation commands in English (the input-language) paired with a desired action sequence (the output-language). For instance, the input-language sentence JUMP TWICE AND RUN LEFT is paired to the output-language actions sequence JUMP JUMP LTURN RUN. The rest of our exposition uses SMALL CAPS to denote examples in the input-language, and LARGE CAPS to denote examples in the output-language. Appendix A contains a full description of the grammar generating the SCAN language.

To evaluate the compositional generalization abilities of sequence-to-sequence (seq2seq) machine translation models (Sutskever et al., 2014; Bahdanau et al., 2015), Lake & Baroni (2018) proposes four main tasks based on SCAN:

1. *Simple task*: data pairs are randomly split into training and test sets. No compositional generalization is required.

2. *Add jump task*: the only command in the training set containing the verb JUMP is the command JUMP. All commands not containing JUMP are in the training set (for instance, RUN TWICE, and WALK RIGHT THRICE AND LOOK LEFT). The test set contains all commands containing JUMP (for instance, JUMP TWICE, and RUN LEFT AND JUMP RIGHT). To succeed in this task, models must learn that JUMP is a verb, and that any verb can be composed with an adverbial number to be repeated a number of times.

3. *Around right task*: the phrase AROUND RIGHT is held out from the training set; however, both AROUND and RIGHT are shown in all other contexts (for example, AROUND LEFT or OPPOSITE RIGHT). To succeed at this task, models must learn that both RIGHT and LEFT are directions, and can be combined with AROUND and OPPOSITE.

4. *Length generalization task*: the training set contains pairs such that the length of the action sequence in the output-language is shorter than 24 actions. The test set contains all pairs with action sequences of a length greater or equal than 24 actions. The type of compositional ability required to succeed at this task is more difficult to sketch out, as we discuss in Section 6.2.

Lake & Baroni (2018) use these four tasks to demonstrate that state-of-the-art seq2seq translation models (Bahdanau et al., 2015) succeed at *Simple task*, but fail at the other three tasks requiring compositional generalization. Convolutional architectures (Dessì & Baroni, 2019) achieve only slightly better performance, and state-of-the-art methods specially developed to address SCAN tasks fall short from the best achievable performance (Russin et al., 2019), or call for substantial amounts of additional supervision (Lake, 2019).

---

[1]Code available at https://github.com/facebookresearch/Permutation-Equivariant-Seq2Seq

In the following, we take a holistic look at the language compositionality problems in SCAN, and highlight their connection to equivariant maps in group theory.

## 3 SCAN COMPOSITIONALITY AS GROUP EQUIVARIANCE

This section puts forward the hypothesis that:

*Models achieving the compositional generalization required in certain SCAN tasks are equivariant with respect to permutation group operations[2] in the input and output languages.*

To unfold the meaning of our hypothesis, we must revisit some basic concepts in group theory. A discrete group $G$ is a set of elements $\{g_1, \ldots, g_{|G|}\}$, equipped with a binary group operation "·" satisfying the four group axioms (closure, associativity, identity, and invertibility). The sequel focuses on *permutation groups* $G$, whose elements are permutations of a set $\mathcal{X}$, and whose binary group operation composes the permutations contained in $G$. The set of all permutations of $\mathcal{X}$ is a group, but not all subsets of permutations of $\mathcal{X}$ satisfy the four group axioms, and therefore they do not form a group. For each element $g \in G$, we define the *group operation* $T_g : \mathcal{X} \to \mathcal{X}$ as the map applying the permutation $g$ to the element $x \in \mathcal{X}$, to obtain $T_g x$. Armed with these definitions, we are ready to introduce the main object of study in this paper: *equivariant maps*.

**Definition 1** (Equivariant map). Let $\mathcal{X}$ and $\mathcal{Y}$ be two sets. Let $G$ be a group whose group operation on $\mathcal{X}$ is denoted by $T_g : \mathcal{X} \to \mathcal{X}$, and whose group operation on $\mathcal{Y}$ is denoted by $T'_g : \mathcal{Y} \to \mathcal{Y}$. Then, $\Phi : \mathcal{X} \to \mathcal{Y}$ is an *equivariant map* if and only if $\Phi(T_g x) = T'_g \Phi(x)$ for all $x \in \mathcal{X}$ and $g \in G$.

The operation groups $(T_g, T'_g)$ defined above operate on entire sequences, an enormous space when we consider those sequences to be language sentences. In the following two definitions, we relax group operations and equivariant maps to operate at a word level.

**Definition 2** (Local group operations). Let $\mathcal{X}$ be a set of sequences (or sentences), where each sequence $x \in \mathcal{X}$ contains elements $x_i \in \mathcal{V}$ from a vocabulary set $\mathcal{V}$, for all $x_i \in x$. Let $G$ be a group with associated group operation $T_g : \mathcal{X} \to \mathcal{X}$. Then, we say that $T_g$ is a local group operation if there exists a group operation $T_{g_w} : \mathcal{V} \times \mathcal{V}$ such that $T_g x = (T_{g_w} x_1, \ldots, T_{g_w} x_{L_x})$ for all $x \in \mathcal{X}$.

When understanding sequences as language sentences, the group operation $T_{g_w}$ would be a permutation of the words from the language vocabulary. Such operation can be implemented in terms of a permutation matrix, a $|\mathcal{V}| \times |\mathcal{V}|$ matrix with zero/one entries where each row and each column sum to one. Finally, we leverage the definition of local group operations to define locally equivariant maps.

**Definition 3** (Locally equivariant map). Let $\mathcal{X}$ and $\mathcal{Y}$ be two sets of sequences. Let $G$ be a group whose group operation on $\mathcal{X}$ is local in its vocabulary, denoted by $T_g : \mathcal{X} \to \mathcal{X}$, and whose group operation on $\mathcal{Y}$ is local in its vocabulary and denoted by $T'_g : \mathcal{Y} \times \mathcal{Y}$. Then, we say that $\Phi : \mathcal{X} \to \mathcal{Y}$ is an equivariant map if and only if $\Phi(T_g x) = T'_g \Phi(x)$ for all $x \in \mathcal{X}$ and $g \in G$.

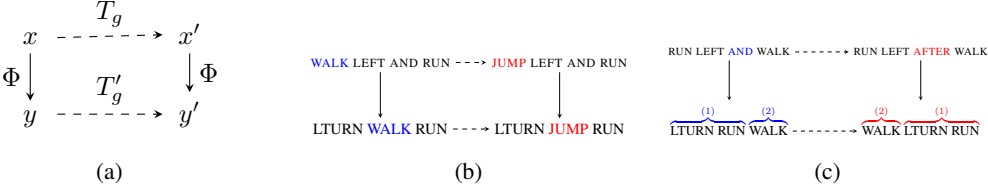

Figure 1: (a) Commutative diagram for equivariance. (b) Local equivariance enables generalization to verb replacement in SCAN. (c) Local equivariance does not enable generalization to conjunction replacement in SCAN.

---

[2]Standard terminology in group theory denotes the elements of a group "to act on" elements of a set. However, as the output language in SCAN is in navigation "actions", we use the "operate" terminology for group elements to avoid ambiguity.

Now, how do equivariances and local equivariances manifest themselves in the world of SCAN? To assist our examples, the commutative diagram in Figure 1a summarizes the group theory notations introduced so far. In Figure 1b and Figure 1c, we parallel these notations to two different examples of compositional skills required to solve SCAN: verb and conjunction replacement. In the SCAN domain, $\mathcal{X}$ is the set of sentences in the input-language, and $\mathcal{Y}$ is the set of sentences in the output-language. Furthermore, let $\Phi$ be a locally equivariant SCAN translation model, and let $G$ be a group with associated local group operations that permutes words in the input- and output- languages.

On the one hand, we observe in Figure 1b that local equivariance enables compositional generalization in the case of verb replacement. This is because replacing one verb in the input-language can be implemented in terms of a local group operation. In turn, this input-verb replacement corresponds deterministically to a second local group operation that replaces the corresponding verb in the output-language. The same would apply to a SCAN task where we are interested in generalizing to the replacement of LEFT and RIGHT. As such, a translation model $\Phi$ with these compositional generalization capabilities must be locally equivariant.

On the other hand, we observe in Figure 1c that local equivariance is insufficient to enable compositional generalization in the case of conjunction replacement. This is because no *local* group operation in the output-language would be able to implement the necessary changes induced by the replacement of AND by AFTER in the input-language. In such cases, we refer to the equivariance as *global equivariance*. In particular, we can see how blocks of multiple words in the output-language swap their relative location. Local equivariances are also insufficient to enable compositional generalization in the *Length generalization* SCAN task and we elaborate on this in Section 6.2.

In the following section, we propose a set of tools to implement equivariant seq2seq translation models, and propose a particular architecture with which we conduct our experiments.

## 4 IMPLEMENTING AN EQUIVARIANT SEQUENCE-TO-SEQUENCE MODEL

We now implement our proposed equivariant seq2seq model, following the encoder-decoder architecture illustrated in Figure 2. Readers unfamiliar with group theory may parse Figure 2 by temporarily discarding the "$G-$" prefixes, and realize that each depicted module is a well-known building block of recurrent neural network models.

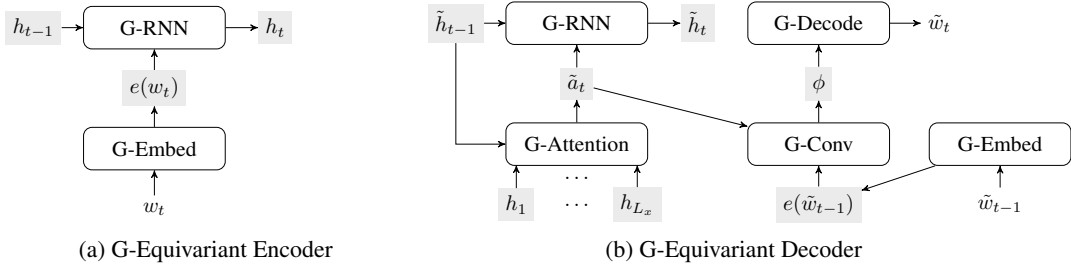

(a) G-Equivariant Encoder          (b) G-Equivariant Decoder

Figure 2: Architecture of our fully-equivariant seq2seq model. Variables shaded in gray are mappings $G \rightarrow \mathbb{R}^K$, implemented as $|G| \times K$ matrices. Encoder and decoder meet at $\tilde{h}_0 := h_{L_x}$.

To make our model equivariant, we will make intense use of *group convolutions*.

**Definition 4** (Group convolution (Kondor & Trivedi, 2018)). Let $G$ be a discrete group. Let $f : G \rightarrow \mathbb{R}^K$ be an input function. Let $\psi = \{\psi^i : G \rightarrow \mathbb{R}^K\}_{i=1}^{K'}$ be a set of learnable filter functions. Then, each scalar real entry from the result of $G$-convolving $f$ and $\psi$ is given by a $|G| \times K'$ matrix with entries

$$G\text{-Conv}(f; \psi)_{g,i} = \sum_{h \in \text{dom}(f)} \sum_{k=1}^{K} f_k(h) \psi_k^i(g^{-1}h), \tag{1}$$

for all $g \in G$ and $i \in \{1, \ldots, K'\}$. As shown by (Kondor & Trivedi, 2018), the $G$-Conv layer is equivariant wrt the operations of $G$. We apply this definition in two ways: (i) "convolving" words with learnable filters to generate equivariant embeddings. Later, when we introduce our notations, we

discuss how words may be viewed as functions so as to fit the definition. And (ii) convolving two group representations, in which case $\mathrm{dom}(f) = G$.

We note that there are several additional methods proposed in the literature for constructing permutation equivariant layers (e.g. , Zaheer et al., 2017; Ravanbakhsh et al., 2017). However, as demonstrated by Kondor & Trivedi (2018); Bloem-Reddy & Teh (2019), the above form is very general and subsumes most alternatives. Further, while layers based on weight-sharing may be more efficient than the general form of Definition 4, the parameter tying restricts the capacity of the layer. For example, the permutation equivariant layer of Zaheer et al. (2017) requires weight matrices that are restricted to a form $\lambda \boldsymbol{I} + \gamma (\boldsymbol{11})^T$, with learnable parameters $\lambda$ and $\gamma$. This layer has fewer learnable parameters than the convolutional form of Definition 4. Thus, for reasons of generality and capacity, we employ the general and expressive convolutional form of Definition 4 for our permutation equivariant layers.

**Equivariant with respect what group?** The previous $G$-Conv layer requires choosing a discrete group $G$. As hinted in Section 3, we will choose $G$ to contain $|G|$ permutations of language vocabularies, e.g. products of cyclic groups on sets of words. Note that for a vocabulary size of $|V|$, the set of all permutations has a size of $n!$. However, it suffices to consider subgroups containing permutations such that every word can be reached by composing elements of the subgroup. For example, while the group of permutations on the four verbs in SCAN consists of 24 elements, it will suffice to choose $G$ as the circular shift group on the four verbs, which is a subgroup of four elements. Following standard notation in group theory, we write $g \cdot h$ to denote the composition of two group elements $g, h \in G$, and $g^{-1}$ to denote the inverse element of $g$.

As final preliminaries, denoting $[V] = \{1, \ldots, |V|\}$, we represent a word $w$ in the input-language by the function $w : [V] \to \{0, 1\}$, where $\sum_{v \in [V]} w(v) = 1$, and similarly by using $\tilde{w} \in \tilde{V}$ for the output-language. These notations are functional representations of word one-hot encodings that will play well with our notations. Note that this representation is equivalent to one-hot vectors, and in what follows we use the shorthand $w$ for the one-hot vector representation of words.

To avoid notational clutter, we use $g$ to denote the permutation-matrix-representation of the corresponding group element. Thus, the group operation on a word $gw$ can be implemented as matrix multiplication between the permutation matrix $g$ and the one-hot vector $w$. Note that this operation results in another one-hot vector, i.e. another word in the vocabulary. Similarly, the binary group operation can be written as matrix multiplication $gh$ between two group members $g, h \in G$. Here too, multiplication of permutation matrices results in permutation matrix, so $gh \in G$.

We now describe each of the components in our $G$-equivariant translation model, by following the transformation process of an input sequence $x = (w_1, \ldots, w_{L_x})$ (in SCAN, a navigation command in English) into its output translation $y = (\tilde{w}_1, \ldots, \tilde{w}_{L_y})$ (in SCAN, a sequence of actions).

## 4.1 G-Equivariant Encoder

Upon arrival, the input-language sentence $x = (w_1, \ldots, w_{L_x})$ is sent to a $G$-equivariant encoder. The first step in the encoding process is to transform each input word $w_t$ into a permutation equivariant embedding $e(w_t)$. As mentioned before, each word $w_t$ is represented by the one-hot vector $w_t : [V] \to \{0, 1\}$. The corresponding embedding is obtained by applying a set of $K$ 1-dimensional learnable filter functions $\{\psi^i : [V] \to \mathbb{R}\}_{i=1}^K$ in a group convolution (throughout the section, we use $K$ everywhere to ease notation). Using Definition 4, the embedding, which we call $G$-Embed, is then represented as a matrix $\mathbb{R}^{|G| \times K}$, where

$$e(w)_{g,i} = G\text{-Embed}(w; \psi)_{g,i} = \psi^i(g^{-1}w), \tag{2}$$

for all $g \in G$ and $i = \{1, \ldots, K\}$. Note that since $w$ is a one-hot vector, $G$-Embed is a particularly simple instantiation of Definition 4, as summation over $\mathrm{dom}(f)$ consists of only a single term. The corresponding embedding is a function $e(w_t) : G \to \mathbb{R}^K$, which can be represented as a $|G| \times K$ matrix, where each row corresponds to the embedding of the word $gw$ for a particular $g \in G$. This layer can be implemented by defining $\psi$ with standard deep learning embedding modules.[3]

---

[3]e.g. PyTorch NN.EMBEDDING

Importantly, we note that for this layer, both $\psi$ and $w$ are functions on $[V]$. However, the resulting embedding $e(w)$ is a function on the group $G$. Therefore, in all subsequent computations we will require the learnable filters $\psi$ to also be functions on $G$.

We illustrate this layer with an example. Let $G$ be the cyclic group that permutes the words LEFT and RIGHT. We can think of $g_1$ as the identity, and $g_2$ as permuting the words LEFT and RIGHT (leaving all other words unchanged). In this case, embedding LEFT results in the $2 \times K$ matrix $[\psi(\text{LEFT})^T, \psi(\text{RIGHT})^T]^T$, while embedding JUMP results in $[\psi(\text{JUMP})^T, \psi(\text{JUMP})^T]^T$, since both $g_1$ and $g_2$ act as the identity permutation for JUMP.

Next, the word embedding $e(w_t)$ is sent to a permutation equivariant Recurrent Neural Network ($G$-RNN). The cells of a $G$-RNN mimic those of a standard RNN, where linear transformations are replaced by $G$-Convs (Definition 4). This cell receives two inputs (the word embedding $e(w_t)$ and the previous hidden state $h_{t-1}$) and returns one output (the current hidden state $h_t$), all three being functions $G \to \mathbb{R}^K$, parametrized as $|G| \times K$ matrices. More specifically:

$$h_t = G\text{-RNN}(e(w_t), h_{t-1}) = \sigma(G\text{-Conv}(h_{t-1}; \psi_h) + G\text{-Conv}(e(w_t); \psi_e)), \tag{3}$$

where $\psi_h, \psi_e \colon G \to \mathbb{R}^K$ are learnable filters (represented as $|G| \times K$ matrices), and $\sigma$ is a point-wise activation function.

The cell $G$-RNN is equivariant because the sum of two equivariant representations is equivariant (Cohen & Welling, 2016), and the pointwise transformation of an equivariant representation is also equivariant. To initialize the hidden state, we set $h_0 = \vec{0}$. We note that our experiments use the equivariant analog of LSTM cells (Hochreiter & Schmidhuber, 1997), which we denote $G$-LSTM, since these achieved the best performance. We include the architecture of $G$-LSTM cells in Appendix B.

This completes the description of our equivariant encoder, illustrated in Figure 2a.

## 4.2    G-EQUIVARIANT DECODER

Once the entire input-language sentence $x = (w_1, \ldots, w_{L_x})$ has been encoded into the hidden representations $h = (h_1, \ldots, h_{L_x})$, we are ready to start the decoding process that will produce the output-language translation $y = (\tilde{w}_1, \ldots, \tilde{w}_{L_y})$.

As illustrated in Figure 2b, our equivariant decoder is also run by an equivariant recurrent cell $G$-RNN. We denote the hidden states of the recurrent decoding process by $\tilde{h}_t$, where $\tilde{h}_0 = h_{L_x}$. At time $t$, the two inputs to the decoding $G$-RNN cell are the previous hidden state $\tilde{h}_{t-1}$ as well as an attention $\bar{a}_t$ over all the encoding hidden states $h$. (Once again, all variables are mappings $G \to \mathbb{R}^K$ implemented as $|G| \times K$ matrices.)

Attention mechanisms (Bahdanau et al., 2015; Vaswani et al., 2017) have emerged as a central tool in language modelling. Fortunately, attention mechanisms are typically implemented as linear combinations, and a linear combination of equivariant representations is itself an equivariant representation. We now leverage this fact to develop an equivariant attention mechanism. Given all the encoder hidden states $h$, as well as the previous decoding hidden state $\tilde{h}_{t-1}$, we propose the equivariant analog of dot-product attention (Luong et al., 2015) as

$$\bar{a}_t = G\text{-Attention}(\tilde{h}_{t-1}, h) = \sum_{j=1}^{L_x} \alpha_{t,j} h_j, \text{ where} \tag{4}$$

$$\alpha_{t,j} = \frac{\exp \beta_{t,j}}{\sum_{k=1}^{L_x} \exp \beta_{t,k}}, \text{ and } \beta_{t,j} = \sum_{g \in G} \tilde{h}_{t-1}(g)^\top h_j(g). \tag{5}$$

Following Figure 2b, the attention $\bar{a}_t$ and a $G$-embedding $e(\tilde{w}_{t-1})$ for the previous output word are concatenated and sent to a $G$-Convolution.[4] The concatenation with $e(\tilde{w}_{t-1})$ provides the decoder with information regarding the previously embedded word. In practice, during training we use teacher-forcing (Williams & Zipser, 1989) to provide the decoder with information about the correct output sequences. This process returns a final hidden representation $\phi : G \to \mathbb{R}^K$.

---

[4]To embed $\tilde{w}_0$ we use a special "start of sentence" symbol, appended to the output language vocabulary.

As a final step in the decoding process, we need to convert $\phi$ into a collection of logits over the output-language vocabulary. Then, sampling from the categorical distribution induced by these logits at time $t$ (or taking the maximimum) will produce the word $\tilde{w}_t$, to be appended in the output-language translation, $y$. This final decoding module can be implemented as follows:

$$G\text{-Decode}(\phi; \psi)_{\tilde{w}} = \sum_{h \in G} \sum_{k=1}^{K} \phi_k(h) \psi_k(h^{-1}\tilde{w}), \tag{6}$$

where $\psi = [\tilde{V}] \to \mathbb{R}^k$ are the learnable parameters of this layer (represented by a $|\tilde{V}| \times K$ matrix).

Recall that $\phi(h) \in \mathbb{R}^K$ is the final-layer representation for the group element $h$, and that $h^{-1}\tilde{w}$ is the inverse element of $h \in G$ applied to the output word $\tilde{w}$ (represented as a one-hot vector), which results in another word in the output language. Thus, $\psi$ is a learnable embedding of the output words into $\mathbb{R}^K$. This layer is evaluated at every $\tilde{w}$ in the output vocabulary to produce a scalar. The resulting vector of logits represents a categorical distribution over the output vocabulary. While similar, this layer is not a group convolution (Definition 4). Rather, equivariance for this module is achieved via parameter-sharing (Ravanbakhsh et al., 2017).

This completes the description of our equivariant decoder, illustrated in Figure 2b. Composing the equivariant encoder and decoder results in our complete sequence-2-sequence model. Importantly, since all operations in this model are equivariant, the complete model is itself also equivariant to the group $G$ (Kondor & Trivedi, 2018). In Section 6, we provide further implementation details for our model, and detail our empirical evaluation of its equivariant properties and their relation to the SCAN tasks described in Section 2.

## 5 RELATED WORK

In this section we review state-of-the-art methods to address SCAN compositional tasks. We focus on two recent models that we will compare to in our experiments.

On the one hand, the syntactic attention model of Russin et al. (2019) builds on the idea that compositional generalization can be achieved by language models given the correct architectural organization. Borrowing inspiration from neuroscience, Russin et al. (2019) argue that compositionality might arise when using separate processing channels for *semantic* and *syntactic* information. In their model, the attention weights depend on a recurrent encoding of the input sequence, which they refer to as the *syntactic* representation. The attention weights are then applied to separate, context-independent embeddings of the words in the input sequence, which intend to model a *semantic* representation. We find (Russin et al., 2019) interesting from a group equivariance perspective, since one way to enforce equivariance is to use an *invariant* representation (about syntax) together with an additional representation (about semantics) that maintains the information about the original "sentence pose".

On the other hand, the meta-learning (Thrun & Pratt, 2012; Schmidhuber, 1987) approach of Lake (2019) is a model that *learns to generalize*. In particular, Lake (2019) designs one specific and complex meta-learning procedure for each SCAN task, where a distribution over tasks is provided to the learner (Finn et al., 2017; Gordon et al., 2018). For example, in the *Add jump* and *Around right* tasks, the meta-learning procedure of Lake (2019) samples permutations from the relevant groups (the permutation groups on the verbs and set of directions, respectively). This is interpreted as data-augmentation, a valid procedure for encouraging equivariance (Cohen & Welling, 2016; Andreas, 2019; Weiler et al., 2018). However, at test-time, Lake (2019) sets the context set to the correct mapping between the permuted commands and their corresponding actions. For example, in the *Add jump* task, the context set for meta-testing would consist of the following pairs: {(WALK, WALK), (RUN, RUN), (LOOK, LOOK), (JUMP, JUMP) }. This is equivalent to providing the model with one-to-one information regarding the correct command-to-action mapping for the permuted words.

## 6 EXPERIMENTS

We now evaluate the empirical performance of our equivariant seq2seq model (described in Section 4) on the four SCAN tasks (described in Section 2). We compare our equivariant seq2seq to regular seq2seq models (Lake & Baroni, 2018), convolutional models (Dessì & Baroni, 2019), the syntactic attention model of Russin et al. (2019), and the meta-learning approach of Lake (2019). The compared seq2seq models use bi-directional, single-layer LSTM cells with 64 hidden units. For the equivariant

| Model | Simple | Add Jump | Around Right | Length |
|---|---|---|---|---|
| seq2seq (Lake & Baroni, 2018) | 99.7 | 1.2 | NA | 13.8 |
| CNN (Dessì & Baroni, 2019) | 100.0 | $69.2 \pm 9.2$ | $56.7 \pm 10.2$ | 0.0 |
| Syntactic Attention (Russin et al., 2019) | 100.0 | $91.0 \pm 27.4$ | $28.9 \pm 34.8$ | $15.2 \pm 0.7$ |
| Meta seq2seq (Lake, 2019) | NA | 99.9 | 99.9* | 16.64 |
| seq2seq (comparable architecture) | 100.0 | $0.0 \pm 0.0$ | $0.02 \pm 2e\text{-}2$ | $12.4 \pm 2.3$ |
| **Equivariant seq2seq (ours)** | 100.0 | $99.1 \pm 0.04$ | $92.0 \pm 0.24$ | $15.9 \pm 3.2$ |

Table 1: Test accuracies for four SCAN tasks, comparing our equivariant seq2seq to the state-of-the-art.

seq2seq models, we use the cyclic permutation group on the verbs for the *Add jump* task, and the cyclic permutation group on directions for the *Around right* task. For *Length*, we use the product of those groups. Our model knows that the same group operates on both the input- and output- languages. However, it does not receive information regarding the correspondence between commands and actions in the set of words being permuted in the input / output languages. This is in contrast to Lake (2019), where (as stated in Section 5), it is necessary to provide the model with explicit information regarding the correct command-to-action mapping at test-time.

Training procedures match those of Lake & Baroni (2018) where possible. We train models for $200k$ iterations, where each iteration consists of a minibatch of size 1, using the Adam optimizer to perform parameter updates with default parameters (Kingma & Ba, 2015) with a learning rate of 1e-4. We use teacher-forcing (Williams & Zipser, 1989) with a ratio of 0.5, and early-stopping based on a validation set consisting on $10\%$ of the training examples. As in previous works, we compute test accuracies by counting how many exact translations each model provides, across the test set associated to each task.

## 6.1 RESULTS

Table 1 summarizes the results of our experiments. First and as expected, all models achieve excellent performance on the *Simple task*, which does not require any form of compositional generalization.

Second, our equivariant seq2seq model performs very well at the *Add jump* and *Around right* SCAN tasks, which are the two tasks satisfying our local equivariance assumption from Definition 3. Our equivariant seq2seq model significantly outperforms the regular seq2seq (Lake & Baroni, 2018) and convolutional (Dessì & Baroni, 2019) models, as well as the state-of-the-art methods of Russin et al. (2019) and Lake (2019). This result is an encouraging piece of evidence supporting our main hypothesis from Section 3. Next, let us compare the results of our equivariant seq2seq model with the previous state-of-the-art Russin et al. (2019); Lake (2019) in more detail.

On the one hand, the syntactic attention model of Russin et al. (2019) achieves significant improvements over baselines methods at the *Add jump* SCAN task. However, it does not fare so well on the *Around right* task. Furthermore, its performance has high variance. Although we here report the numbers from Russin et al. (2019), we observed such high variance in our own implementation as well, where the model often achieved $0\%$ test accuracy. We hypothesize that modeling the invariance of the syntactic attention directly would result in improved performance and stability. This can be achieved, for instance, by replacing all verbs in the syntactic module by a shared word. As expected, by explicitly exploiting equivariance, our model outperforms Russin et al. (2019) on the *Add jump* and *Around right* SCAN tasks, also being much more robust.

On the other hand, the meta-learning model of Lake (2019) achieves excellent performance on the local equivariance tasks *Add jump* and *Around right*. This is additional evidence supporting the usefulness of local equivariance. In contrast to our model, Lake (2019) requires (i) a complicated model and training procedure tailored to each task, (ii) providing the model with the correct permutation of words, equivalent to telling the model the "true" mappings between the input and output words, and (iii) augmenting the set of words being permuted, to ensure enough diversity in the training distribution (for instance, adding additional directions beyond RIGHT and LEFT).

## 6.2 On the difficulty of length generalization

As seen in Table 1, length generalization remains a tough challenge in SCAN. While generating long sequences is a known challenge in seq2seq models (Bahdanau et al., 2015), we believe that this is not the main issue with our equivariant seq2seq model, as it is able to produce long translations when these appear in the training set (as are the other models). Therefore, this is not a capacity problem, but one of not being able to express the *Length generalization* SCAN task in terms of *local* equivariances on both input- and output- languages. We hypothesize that this is the very reason why (Russin et al., 2019; Lake, 2019) also fail on this task.

However, we suspect that some forms of *local* equivariance on the input language, but *global* equivariance on the output language, may help. For example, RUN TWICE, RUN THRICE and RUN AROUND LEFT TWICE are all input commands contained in the training set of the length task. A trained seq2seq model is able to execute them, but fails on the unseen test command RUN AROUND LEFT THRICE, suggesting that the network did not correctly understand the relationship between TWICE and THRICE. Using a network that is explicitly equivariant to the permutation of TWICE and THRICE should be able to generalize correctly on RUN AROUND LEFT THRICE. However, while the TWICE-THRICE permutation is a *local* group operation (Definition 2), the corresponding operation on the output language, which is to repeat the same action sequence multiple times, is a *global* group operation. Similarly, permuting AND and AFTER in the input sequence using a local group operation, while operating globally on the output language by permuting the order of the associated actions, should help succeed on the *Length generalization* SCAN task. How to formalize the aforementioned global operations on the output language and build the desired equivariant network remains a fascinating open research question that we leave for future work.

## 7 Discussion and Future Work

This work has introduced hypothesis linking between *group equivariance* and *compositional generalization* in language. Motivated by this hypothesis, we have proposed an equivariant seq2seq translation model, which achieves state-of-the-art performance on a variety of SCAN tasks.

Our work has several points for improvement. Most importantly, our model requires knowing the permutation symmetries of interest, to be provided by some domain expert. While this is simple to do in the synthetic language of SCAN, it may prove more difficult in real-world tasks. We propose three directions to attack this problem. (i) Group words by their *parts-of-speech* (e.g., nouns, verbs, etc.), which can be done automatically by standard part-of-speech taggers (Màrquez & Rodríguez, 1998); (ii) *Learn* such groupings of words from corpora, for example using the recent work of Andreas (2019); (iii) Most appealingly, parameterize the symmetry group and learn operations end-to-end while enforcing the group structure. For permutation symmetries, the group elements can be parameterized by permutation matrices, and learned from data (Lyu et al., 2019). Our preliminary work in this direction hints that this is a fruitful avenue for future research.

A further consideration to address is that of computational overhead. In particular, for the convolutional form we use in this work (Definition 4), computational complexity scales linearly with the size of the group, $\mathcal{O}(|G|)$. This arises from the need to sum over group elements when the representation is a function on $G$, and may be prohibitive when considering large groups. One way of addressing this issue when large symmetry groups are of interest is to consider more efficient computational layers for permutation equivariance (e.g Zaheer et al., 2017; Ravanbakhsh et al., 2017). These methods incur less computational overhead at the cost of restricting the layer capacity. Another interesting option for future research is to consider sub-sampling group elements when performing the summation in Definition 4, which requires further consideration of the consequences of doing so.

Another exciting direction for future research is to consider global equivariances. Many operations of interest, e.g. groups operating directly on parse trees, can only be expressed as global equivariances. Modeling these equivariances holds exciting possibilities for capturing non-trivial symmetries in language tasks, but also requires more sophisticated machinery than is proposed in this work.

Finally, in further theoretical work, we would like to explore the relation between our equivariance framework and the idea of compositionality in formal semantics (Kratzer & Heim, 1998). On the one hand, the classic idea of compositionality as an isomorphism between syntax and semantics is intuitively related to the notion of group equivariance. On the other hand, as shown by the failures

at the length generalization example, it is still unclear how to apply our ideas to more sophisticated forms of permutation, such as those involving grammatical phrases rather than words. This would also require to extend our approach to account for the *context-sensitivity* that pervades linguistic composition (c.f., the natural interpretation of "run" in "run the marathon" vs. "run the code").

ACKNOWLEDGMENTS

We thank Emmanuel Dupoux and Clara Vania for helpful feedback and discussions.

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

## A    DETAILS ON THE SCAN DATASET

SCAN is composed from a non-recursive grammar, as shown in Figure 3. In particular, SCAN consists of all commands that can be generated from this grammar (20,910 command sequences), with their deterministic mapping into actions, as detailed by Figure 4

$$
\begin{array}{lll}
C \to S \text{ and } S & V \to D[1] \text{ opposite } D[2] & D \to \text{turn left} \\
C \to S \text{ after } S & V \to D[1] \text{ around } D[2] & D \to \text{turn right} \\
C \to S & V \to D & U \to \text{walk} \\
S \to V \text{ twice} & V \to U & U \to \text{look} \\
S \to V \text{ thrice} & D \to U \text{ left} & U \to \text{run} \\
S \to V & D \to U \text{ right} & U \to \text{jump}
\end{array}
$$

Figure 3: The grammar used to generate commands in the SCAN domain. Indexing notation is used to allow infixing: read $D[i]$ as "the $i$-th element directly dominated by category $D$". Image borrowed from Lake & Baroni (2018).

$$
\begin{array}{ll}
\llbracket \text{walk } \rrbracket = \text{WALK} & \llbracket u \text{ opposite left} \rrbracket = \llbracket \text{turn opposite left} \rrbracket \ \llbracket u \rrbracket \\
\llbracket \text{look} \rrbracket = \text{LOOK} & \llbracket u \text{ opposite right} \rrbracket = \llbracket \text{turn opposite right} \rrbracket \ \llbracket u \rrbracket \\
\llbracket \text{run} \rrbracket = \text{RUN} & \llbracket \text{turn around left} \rrbracket = \text{LTURN LTURN LTURN LTURN} \\
\llbracket \text{jump} \rrbracket = \text{JUMP} & \llbracket \text{turn around right} \rrbracket = \text{RTURN RTURN RTURN RTURN} \\
\llbracket \text{turn left} \rrbracket = \text{LTURN} & \llbracket u \text{ around left} \rrbracket = \text{LTURN } \llbracket u \rrbracket \text{ LTURN } \llbracket u \rrbracket \text{ LTURN } \llbracket u \rrbracket \text{ LTURN } \llbracket u \rrbracket \\
\llbracket \text{turn right} \rrbracket = \text{RTURN} & \llbracket u \text{ around right} \rrbracket = \text{RTURN } \llbracket u \rrbracket \text{ RTURN } \llbracket u \rrbracket \text{ RTURN } \llbracket u \rrbracket \text{ RTURN } \llbracket u \rrbracket \\
\llbracket u \text{ left} \rrbracket = \text{LTURN } \llbracket u \rrbracket & \llbracket x \text{ twice} \rrbracket = \llbracket x \rrbracket \ \llbracket x \rrbracket \\
\llbracket u \text{ right} \rrbracket = \text{RTURN } \llbracket u \rrbracket & \llbracket x \text{ thrice} \rrbracket = \llbracket x \rrbracket \ \llbracket x \rrbracket \ \llbracket x \rrbracket \\
\llbracket \text{turn opposite left} \rrbracket = \text{LTURN LTURN} & \llbracket x_1 \text{ and } x_2 \rrbracket = \llbracket x_1 \rrbracket \ \llbracket x_2 \rrbracket \\
\llbracket \text{turn opposite right} \rrbracket = \text{RTURN RTURN} & \llbracket x_1 \text{ after } x_2 \rrbracket = \llbracket x_2 \rrbracket \ \llbracket x_1 \rrbracket
\end{array}
$$

Figure 4: The SCAN translation mapping. Double brackets denote the interpretation function translating SCAN's command (input language) into the action (output) language (which are denoted by upper-case strings. Image borrowed from Lake & Baroni (2018).

## B    G-LSTM DETAILS

We provide the equations for implementing our G-LSTM. Given $\boldsymbol{h}_{t-1}, \boldsymbol{c}_{t-1}$ (hidden state and cell-state, respectively), and $e(w)_t$ (all of which are of the form $G \mapsto \mathbb{R}^K$, we can describe the G-LSTM cell as follows:

$$
\begin{aligned}
\boldsymbol{i}_t &= \sigma\left(\boldsymbol{x}_t * \boldsymbol{\psi}_{ii} + \boldsymbol{s}_{t-1} * \boldsymbol{\psi}_{ih}\right); & \boldsymbol{f}_t &= \sigma\left(\boldsymbol{x}_t * \boldsymbol{\psi}_{fi} + \boldsymbol{s}_{t-1} * \boldsymbol{\psi}_{fh}\right) \\
\boldsymbol{g}_t &= \tanh\left(\boldsymbol{x}_t * \boldsymbol{\psi}_{gi} + \boldsymbol{s}_{t-1} * \boldsymbol{\psi}_{gh}\right); & \boldsymbol{o}_t &= \sigma\left(\boldsymbol{x}_t * \boldsymbol{\psi}_{oi} + \boldsymbol{s}_{t-1} * \boldsymbol{\psi}_{oh}\right) \\
\boldsymbol{c}_t &= \boldsymbol{f}_t \circ \boldsymbol{c}_{t-1} + \boldsymbol{i}_t \circ \boldsymbol{g}_t; & \boldsymbol{h}_t &= \boldsymbol{o}_t \circ \tanh(\boldsymbol{c}_t),
\end{aligned}
$$

where $\{\boldsymbol{\psi}_{jk} : G \mapsto \mathbb{R}^K; j \in \{i, f, g, o\}; k \in \{i, h\}\}$ are the learnable filters of the cell. Here we have used the shorthand

$$
\boldsymbol{f} * \boldsymbol{\psi} := [\boldsymbol{f} * \boldsymbol{\psi}](g) \quad \forall g \in G
$$

for two functions on the group.

