# OpenReview forum: "Permutation Equivariant Models for Compositional Generalization in Language"
_ICLR.cc/2020/Conference — Accept (Poster)_

### Official Review · AnonReviewer1 · 2019-10-23
**Official Blind Review #1**

**Rating:** 6

**Review:**

This work focuses on learning equivariant representations and functions over input/output words for the purposes of SCAN task. Basically, the focus is on local equivariances (over vocabulary) such that the effect of replacing and action verb like RUN in the input with the verb JUMP causes a similar change in the output. However, effects requiring global equivariances like learning relationship between "twice" and "thrice", or learning relationships between different kinds of conjunctions are not handled in this work. For learning equivariant functions over vocabulary, group convolutions are used at each step over vocabulary items in both the sequence encoder and decoder.  The results on SCAN task are impressive for verb replacement based experiments and improve over other relevant baselines. Also, improvement is shown on another word replacement task ("around right"), which requires learning corresponding substitutions in output based on the word changes in the input. As expected, for experiments that require global equivariances or no equivariance (simple, length), the difference ion performance is not very pronounced over other baselines.
While this paper does show that modelling effects of word substitution can be handled by the locally equivariant functions, it still cannot account for more complex generalization phenomena which are likely to be much more prevalent especially for domains dealing with natural language that are other than SCAN. Therefor, I think the applicability of the proposed equivariant architectures is rather limited if interesting.

**Experience Assessment:**

I have read many papers in this area.

**Review Assessment: Checking Correctness Of Derivations And Theory:**

I assessed the sensibility of the derivations and theory.

**Review Assessment: Checking Correctness Of Experiments:**

I carefully checked the experiments.

**Review Assessment: Thoroughness In Paper Reading:**

I read the paper thoroughly.

---

> ### Author Response · Authors · 2019-11-14
> **Response to reivew**
>
> We thank the reviewer for their time and effort in reviewing our paper. We are excited that you found our experimental results “impressive”.
>
> We completely agree with your comment: capturing global equivariances in language is an important, interesting, and challenging problem. Despite not having addressed global equivariance in this work, we believe that modelling local equivariances in language is an important first step, and our work provides a proof of concept for this idea, as well as an important link between equivariance and several forms of generalization that are of broad interest to the domain of modelling language. In the future, we certainly intend to expand the investigation to global equivariances as well.

---

### Official Review · AnonReviewer3 · 2019-10-23
**Official Blind Review #3**

**Rating:** 6

**Review:**

The paper presents an architecture that captures equivariance to certain transformations that happen in text, like synonym words and some simple transformation over word order.

* General comments:

Increasing compositional generalization using equivariance is a very interesting idea. Sections 1-3 are well written and the solution of modeling the translation function as a G-equivariant function is well motivated.

Section 4 is far less clear. In its current form, it is very hard to understand the model construction as well as the design choices. This section should be significantly improved in order for me to increase my score. A direct by-product of the confusing writing is that the experiments cannot  be reproduced.

The experiments show improvement in one out of four tasks, where the single phrase “Around right” is held from the training set. There are no examples, not qualitative analysis, no ablation experiments. Overall, more evidence needed to convince that the approach is useful. In addition to deeper error analysis, the authors can hold out other phrases (e.g., “around left”, and many others).

* Specific comments which I hope the authors address:

1. To the best of my understanding, the authors do not explicitly specify the group G that they want to be invariant to. Is it a product of a few cyclic groups? (a cycle for each set of words that are interchangeable?)

2. The authors suggest using G-convolution, i.e. the group convolution on G. This is in contrast to the (arguably) more popular choice of using linear layers that are G-equivariant (as in, for example,  deep sets (Zaheer et al. 2017), Deep Models of Interactions Across Sets (Hartford et al. 2018),Universal invariant and equivariant graph neural networks (Keriven and Peyré ) and in general convolutional layers for learning images).
I have several questions regarding this choice:
2a. Can the authors discuss the differences/advantages of this approach over the approach mentioned above? It seems like the approach mentioned above will be more efficient (as there is no need to sum over all group elements)
2b. In order to use G-convolution, one has to use functions defined on G. Can the authors explain how they look on the input words as functions on G?
2c. How is the G-Conv actually implemented?
2d. Can the authors provide some intuition to what this operator does?

3. Is the whole model G-equivariant? The authors might want to clearly state this. To the best of my understanding, this is the main motivation of this construction.

4. It might be helpful for readers that are not familiar with deep learning for NLP tasks to provide a visualization of the full model (can be added to the appendix)

5. Why are words represented as infinite one-hot sequences? Don’t we assume a finite vocabulary? This is pretty confusing.

6. As a part of the G-RNN the authors apply a G-conv to the state h_{t-1}. What is the dimension of this hidden state? How does G act on it?

7. Please explicitly state the dimensions of each input/output/parameter in the network (this can be combined with the illustration above illustration)

* Minor comments:

Section 4.1 pointwise activation are in general equivariant only to permutation representations
Page 2 - typo - ‘all short’-> ‘fall short’


**Experience Assessment:**

I have read many papers in this area.

**Review Assessment: Checking Correctness Of Derivations And Theory:**

I assessed the sensibility of the derivations and theory.

**Review Assessment: Checking Correctness Of Experiments:**

I assessed the sensibility of the experiments.

**Review Assessment: Thoroughness In Paper Reading:**

I read the paper at least twice and used my best judgement in assessing the paper.

---

> ### Author Response · Authors · 2019-11-14
> **Response to review (1 / 2)**
>
> We thank the reviewer for his time and effort in providing the detailed review of our work. We are grateful that you found the core idea of the paper well-motivated and interesting, and found most sections of the paper to be well-written. We address your comments in order below.
>
> Major comments
>
> R3.1: Section 4 is far less clear.
>
> We agree that the writing of section 4 requires improvement. We have put significant effort into revising this section in the manuscript, focusing on the comments and suggestions that you and the other reviewers made. We believe that the section is now easier to follow. Further, we appreciate your concerns regarding the reproducibility of the model and experiments. Along with improving the clarity of the writing, we would like to state that code will be made available to reproduce all of our models and experiments.
>
> R3.2: In addition to deeper error analysis, the authors can hold out other phrases (e.g., “around left”, and many others).
>
> We chose to focus on tasks for which baseline models’ performance was available from the related literature, without which it is difficult to judge the usefulness of our proposed model. We note that due to the symmetry in the data generation process, there is not a meaningful difference between holding out “around right” and “around left”, and similarly for the verbs.
>
> Regarding an ablation study, our standard seq-2-seq model with a matched architecture is meant to play this role, i.e. ablating the use of permutation equivariance. We are happy to provide further ablations that we may not have considered -- is there something specific you were thinking of?
>
> Finally, regarding error analysis: the key purpose of the experiments is to validate our hypothesis that permutation equivariance as we have defined it leads to the improved performance in tasks requiring the type of compositional generalization required by the SCAN tasks. We believe that our experiments provide this evidence. Detailed analysis of the errors made by sequence-2-sequence models in this setting is explored in considerable depth in [4]. Regardless, we agree that further analysis of the errors made by our model may be of interest. However, relevant analyses of this form require some thought, and we are reluctant to publish analyses at very short notice. We will put in thought and effort to relevant analyses, and will provide these for future versions of the manuscript.
>
> R3.1: Explicitly specifying the group G.
>
> You are correct in stating that the group G may be a product of several cyclic groups, one for each set of words. In particular, in the experimental section we state what group are used for each experiment ($\sS_4$ over verbs for “Add Jump”, $\sS_2$ over directions for “Around Right”, and their product group for “Length”), and we have made this statement more explicit in the revision.
>
> R3.2: Additional options for equivariant layer implementations.
>
> Thank you for pointing this out — we agree that this is an important discussion missing from the submitted manuscript. We address you points in order:
> 	a. You are correct in stating that there are additional options for this layer. However, as discussed in [1,2], the form we use is the most general, and many of the methods you mention can be seen as special cases of the convolutional form [1]. Further, while the form provided by Zaheer et al. 2017 is more efficient (as you say, it relieves the need to sum over group elements), it is also far more restrictive. Examining Lemma 3 and eq. 4 of that paper, one can see that their proposed layer has only 2 free parameters (for the 1d case), and relies on extremely restrictive parameter sharing to achieve permutation equivariance. Thus, both for generality and expressivity of the layers, we opted for the convolutional form.
> 	b. As discussed in [3], we can apply a G-Conv to a function on an input domain which is acted on by the group elements (see e.g. Eq. 10 of [3], and Definition 4 in our manuscript). To this end, we define words as functions from indices (integers) to {0,1} (intuitively, 1-hot vectors), which are acted on by the elements in G. This allows us to fully define the G-Conv operator for words.

---

> > ### Author Response · Authors · 2019-11-14
> > **Response to review (2 / 2)**
> >
> > c. The G-Conv is implemented differently for different uses. We can represent words as 1-hot vectors, and elements of $g$ as permutation matrices, and apply $gw$ (to a word), $gh$ (the binary group operation) using standard matrix multiplication. With this in place, when used to embed words, we can implement the G-Conv by inheriting regular word embeddings, and letting the word embedding be the embedding of the word after each group element is applied. For convolving two representations on the group, we can similarly use the binary group operation, and inherit standard convolutional operations to implement the convolutional form in definition 4. We have added such discussions to Section 4 to help understand these operations. Further, full PyTorch code will be released following the review process implementing all of these operations.
> > 	d. As mentioned in our response to R2, the G-Conv “keeps track” of the representation of the word under each element of the group, by stacking these representations in a matrix. This allows the model to share parameters across words in a set, while keeping track of which word gave rise to each representation.
> >
> > R3.3: Is the whole model G-equivariant?
> >
> > We thank you for raising this issue. Yes, the complete model is G-equivariant, and this is extremely important to the crux of our argument. As discussed in [2], stacking equivariant computations is itself equivariant. Therefore, since our model is composed only of equivariant operations, the complete model is itself equivariant. We have added an explicit statement to this end in the revised manuscript.
> >
> > R3.4: Provide a visualization of the full model.
> >
> > We thank you for this suggestion. This was our intention in providing Figure 2 in the manuscript, though we agree with you that a more detailed visualization could be useful to some readers. For the current revision, we have not been able to provide such a visualization. However, if the manuscript is accepted we will work on providing better visualizations of the model for the final version of the paper.
> >
> > R3.5: Why are words represented as infinite 1-hot sequences?
> >
> > This is general notation that was useful for our derivations. However, we agree that this may be confusing to many readers. As such, we have taken your advice and changed the notation such that the vocabularies are strictly finite, and words are represented as 1-hot vectors rather than sequences. Thank you for pointing this out.
> >
> > R3.6: What is the dimension of of the hidden state in G-RNN? How does G act on it?
> >
> > The hidden state in the G-RNN is a representation on the group G, such that it is represented as a matrix of size $|G| \times \R^k$. Here, $k$ is a hyper-parameter that is analogous to the number of hidden units in a standard RNN (in our experiments, we set $k=64$). $G$ acts on the hidden state by permuting the rows of the matrix, which is implemented as matrix multiplication with the permutation matrix representation of the elements of $g \in G$. We have added clarifications and specifications of dimensions throughout section 4.
> >
> >
> > [1] B. Bloem-Reddy, and Y. W. Teh. Probabilistic symmetry and invariant neural networks. 2019.
> > [2] R. Kondor, and S. Trivedi. On the generalization of equivariance and convolution in neural networks to the action of compact groups. 2018.
> > [3] T. S. Cohen, and M. Welling. Group equivariant convolutional networks. 2016.
> > [4] B. M. Lake, and M. Baroni. Generalization without systematicity: on the compositional skills of sequence-to-sequence recurrent networks. 2017.

---

### Official Review · AnonReviewer2 · 2019-10-24
**Official Blind Review #2**

**Rating:** 8

**Review:**


Summary
---

(motivation)
Consider SCAN, a synthetic task where setences like S1="jump twice and run left" are supposed to be translated into action sequences like A1=JUMP JUMP LTURN RUN. One might replace the word "jump" in S1 with "walk" then translate to get A2=WALK WALK LTURN RUN. If instead S1 is translated into A1 and then the action JUMP is replaced with the action WALK then we should still get the same A2. Such a translation model is equivariant to permutations of "jump" and "walk".

This paper aims to
1) define a general notion of compositionality as equivariance,
2) build a model which is compositional in this general sense, and
3) apply the model to SCAN.

(approach - theory)
This work considers this kind of compositionality as equivariance to group actions. Previous work (Kondor & Trivedi, 2018) viewed convolution as equivariance to actions by translation groups. This work views language compositionality as equivariance to actions by permutation groups applied to a set of similar words (e.g. verbs in SCAN).

(approach - model)
The paper proposes G-Embed, G-RNN, G-Attention, and G-Conv (not new) layers that are equivariant to word permuatations (e.g., switching "jump" and "walk"). It then composes these modules in a fairly standard fashion to build a new G-seq2seq model which is invariant to group actions.

(experiments)
Experiments apply a G-seq2seq model to the SCAN tasks, comparing to strong baselines. G-seq2seq requires slightly more knowledge (a set of related words like verbs) than all the baselines, but less knowledge than Lake 2019.
1. G-seq2seq outperforms all baselines except Lake 2019 (unfair comparison) on basic compositional tasks ("Add Jump" and "Around Right").
2. Like other models, G-seq2seq fails on the "Length" task, though it is still among the best performers.


Strengths
---

The theory of compositionality as invariance to actions by permutation groups is new, interesting, and could turn out to be significant.

The proposed models are also new, interesting, and could be significant.

Experiments on SCAN verify that the proposed models work about as expected, sometimes beating strong baselines in the process.


Weaknesses
---

It's hard to know what the impact of this paper will be because 1) it's unclear whether this model can generalize to more useful domains and 2) the presentation may turn some readers away. While neither of these issues can really be solved, I think paper could be substantially better in both aspects. Corresponding suggestions:
1) How expensive is this? It seems quite expensive because the representation size scales with the number of permutation of the set of words equivariance is with respect to. How will it scale to larger problems in terms of computation/memory costs (especially larger vocab sizes)? What knowledge is required for applying this method to new domains--i.e., how do I choose a set of permutation equivariant verbs in general? More discussion of these issues may help increase the impact of the paper.
2) See next section.


Presentation Weaknesses / Points of Confusion / Missing Details
---

To mimic a typical decoder RNN there should be another input which copies the word \tilde{w}_{t-1} from the previous iteration as input, somehow fused with the attention feature \tilde{a}_t. How does the G-RNN know what the last word it generated was?

The notation $g^{-1} w$ in the first equation of section 4.1: I think $\psi^i$ is supposed to take an integer as input but $g^{-1} w$ is a permutation applied to a function. I'm not sure how to apply permutations to functions like w and it doesn't seem like the output should be an integer in any case so I find this notation confusing.

Taking a step back, I find some of the notation (e.g., previous point) a bit confusing. This makes it hard for me to get the main point. I think the idea is that equivariant models can be achieved by tracking a representation (e.g., via rows of the G-Embed matrix) for (almost?) every member of the acting group.

It may help the presentation to more frequently demonstrate the general concepts with examples, though doing so may be in conflict with the general nature of the paper's theoretical contribution. I'm sure this is a familiar tradeoff, but from my perspective the paper would probably be more impactful if the presentation leaned more on examples.

Equation numbers would be a really great addition. I found it hard to reference some of the material in writing my review.

"and the use of algebraic computation"
* This seems specific to the chosen example whereas the rest of the sentence is trying to be general.


Suggestions
---

* This seems related to [1], which uses group theory to define a notion of disentangled representation similar to compositionality. That may inspire future work and would be useful to mention in the related work.

* Why didn't performance on SCAN get to 100%? It would be useful to spend some time addressing points of failure for the model other than compositionality.

* The G-RNN doesn't have a bias. It's not necessary, but it may be interesting to describe why this design choice was made.


[1]: Higgins, Irina et al. “Towards a Definition of Disentangled Representations.” ArXiv abs/1812.02230 (2018): n. pag.


Preliminary Evaluation
---

Quality: The theoretical contributions make sense and the experiments show they lead to useful models.
Clarity: The technical parts of the paper are somewhat unclear, but the rest of the paper is well written.
Significance: As discussed in the Weaknesses section this could turn out to be very significant or not significant at all, but that's true for a lot of good research.
Originality: The general notion of equivariant neural networks and good performance on SCAN are novel.

Overall, this is a very clear accept.

Post-Rebuttal Update
---

There was a lot of agreement between reviewers, though we came to slightly different conclusions about ratings. Though there is significant uncertainty about the impact of this work, I still think 8: Accept is the most appropriate rating. Overall, the other reviews and author responses only increased my confidence that this paper should be accepted.

**Experience Assessment:**

I have published one or two papers in this area.

**Review Assessment: Checking Correctness Of Derivations And Theory:**

I assessed the sensibility of the derivations and theory.

**Review Assessment: Checking Correctness Of Experiments:**

I assessed the sensibility of the experiments.

**Review Assessment: Thoroughness In Paper Reading:**

I read the paper at least twice and used my best judgement in assessing the paper.

---

> ### Author Response · Authors · 2019-11-14
> **Response to review**
>
> We would like to thank the reviewer for a kind and helpful review and useful comments which we believe will significantly improve the paper. We are grateful that you have recognised the novelty in our work and are happy that you find the ideas interesting. Further, your major and minor comments are well-made, and we address these below.
>
> A major comment, also shared by R3, is on the density and difficulty of Section 4. To this end, we have put considerable effort into revising Section 4, and have taken your advice to add examples where possible to help clarify certain concepts. Below we address your more specific comments (which we have also addressed in the revised manuscript).
>
> Major comments
>
> R2.1: How expensive is the method?
>
> This is an important issue, and we thank you for raising it. Generally speaking, the approach scales linearly with the size of the acting group $|G|$. While this may pose an issue, we believe there are several ways to improve this computational issue, e.g., by using less expressive layers for permutation equivariance that do not require summation over the elements of G. In scaling the notion of equivariance to natural language, this is an important issue that must be considered. We have added a discussion on this point to Section 7.
>
> R2.2: How does the G-RNN know what the last word it generated was?
>
> In the decoder model, the output of the G-attention mechanism ($\tilde{a}_t$) is combined (via concatenation) with the G-embedding of the last word ($e(\tilde{w}_{t-1}$). The resulting variable is then passed through a G-convolution before being processed by the G-decoder. This can be seen in Figure 2.b, and is stated on page 6 of the paper (first paragraph on the page). We agree that needs to be made more explicit, and have added wording to the section stating this more clearly.
>
> R2.3: The notation $g^{-1}w$ in the first equation of section 4.1.
>
> $\psi^i$ operates on integers, or equivalently (as in our implementations), on 1-hot vectors. In order to consider our words as functions (so that we may properly define notions of convolutions on words), we represent one-hot vectors as functions from indices to $\{0,1\}$. With the representation of words $w$ as one-hot vectors, we can represent elements $g \in G$ as permutation matrices, in which case $g^{-1}w$ results in another one-hot vector (which can then be passed to $\psi^i$).  We have reworked some of the notation, and added an explicit statement of this nature to section 4, which we hope clarifies the issue.
>
> R2.4: Some of the notation is confusing. This makes it hard for me to get the main point.
>
> We agree with your point on the notation, and have reworked large parts in section 4 to make things clearer. We believe that this makes the main points easier to follow. You are correct in saying that, intuitively, the equivariance is achieved by “keeping track” of a representation in the G-matrices for every member of the acting group. To this end, we have also added some examples and intuition to Section 4.
>
>
> Minor comments
>
> R2.5: Equation numbers.
>
> We have added equation numbers to assist in the reviewing.
>
> R2.6: This seems related to Higgins et al., 2018.
>
> Thank you for pointing this reference out to us — it is indeed related and interesting, and we have added a discussion on it to the paper.
>
> R2.7: Why didn’t performance on SCAN go to 100%?
>
> This is an interesting question. We have not focused on this issue as there may be factors unrelated to the equivariance (which is our main focus) that may influence this. [1] provides a very detailed analysis of the performance of sequence-to-sequence models on the SCAN dataset, and many of those analyses carry over to our experiments.
>
> [1] B. M. Lake, and M. Baroni. Generalization without systematicity: on the compositional skills of sequence-to-sequence recurrent networks. 2017.

---

### Author Response · Authors · 2019-11-14
**Response to reviewers**

We thank the reviewers for their detailed reviews, and many helpful comments. We have now uploaded a revised version of the manuscript, reflecting the suggestions. We address specific comments of each reviewer separately, as responses to their reviews. The main revisions and efforts have gone in to improving the clarity of section 4, which we agree is quite dense, and not easy to follow. To this end, we have
	1. improved the notation, taking the advice of the reviewers on several points.
	2. included wording to clarify the dimensions and representations of objects and variables.
	3. included more examples and intuition regarding the proposed computations and models.
We have also added a discussion regarding the computational complexity of our model, as well as addressed several other points raised by the reviewers.
We believe these changes have improved the quality of the paper, and will lead to greater impact. We thank the reviewers for their time and highly useful feedback.

---

### Decision · Program_Chairs · 2019-12-19

**Decision:**

Accept (Poster)

**Comment:**

This paper proposes an equivariant sequence-to-sequence model for dealing with compositionality of language. They show these models are better at SCAN tasks.

Reviewers expressed two major concerns:
1) Limited clarity of section 4 which makes the paper difficult to understand.
2) Whether this could generalize to more complex types of compositionality.

Authors responded by revising Section 4 and answering the question of generalization. While the reviewers are not 100% satisfied, they agree there is enough novel contribution in this paper.

I thank the authors for submitting and look forward to seeing a clearer revision in the conference.